# *In Vitro* Evaluation Reveals Effect and Mechanism of Artemether against *Toxoplasma gondii*

**DOI:** 10.3390/metabo13040476

**Published:** 2023-03-27

**Authors:** Qiong Xu, Yin-Yan Duan, Ming Pan, Qi-Wang Jin, Jian-Ping Tao, Si-Yang Huang

**Affiliations:** 1Institute of Comparative Medicine, College of Veterinary Medicine, Yangzhou University, Jiangsu Co-Innovation Center for Prevention and Control of Important Animal Infectious Diseases and Zoonosis, and Jiangsu Key Laboratory of Zoonosis, Yangzhou 225009, China; 2Joint International Research Laboratory of Agriculture and Agri-Product Safety, The Ministry of Education of China, Yangzhou University, Yangzhou 225009, China

**Keywords:** *Toxoplasma gondii*, artemether, mitochondria, mechanism, *in vitro*

## Abstract

Due to the limited effectiveness of existing drugs for the treatment of toxoplasmosis, there is a dire need for the discovery of new therapeutic options. Artemether is an important drug for malaria and several studies have indicated that it also exhibits anti-*T. gondii* activity. However, its specific effect and mechanisms are still not clear. To elucidate its specific role and potential mechanism, we first evaluated its cytotoxicity and anti-*Toxoplasma* effect on human foreskin fibroblast cells, and then analyzed its inhibitory activity during *T. gondii* invasion and intracellular proliferation. Finally, we examined its effect on mitochondrial membrane potential and reactive oxygen species (ROS) in *T. gondii*. The CC_50_ value of artemether was found to be 866.4 μM, and IC_50_ was 9.035 μM. It exhibited anti-*T. gondii* activity and inhibited the growth of *T. gondii* in a dose-dependent manner. We also found that the inhibition occurred primarily in intracellular proliferation, achieved by reducing the mitochondrial membrane integrity of *T. gondii* and stimulating ROS production. These findings suggest that the mechanism of artemether against *T. gondii* is related to a change in the mitochondrial membrane and the increase in ROS production, which may provide a theoretical basis for optimizing artemether derivatives and further improving their anti-*Toxoplasma* efficacy.

## 1. Introduction

*Toxoplasma gondii* is an obligate intracellular parasite that infects a wide range of warm-blooded mammals, including human beings [1]. About one in three people is infected by *T. gondii* in the world [2]. The infection is asymptomatic or mild for most immunocompetent individuals. For immunodeficiency patients, however, it may cause serious health problems such as encephalitis and retinochoroiditis [3].

The standard therapy for toxoplasmosis relies on a combination of sulfadiazine and pyrimethamine [4]. Both drugs inhibit *T. gondii* by inhibiting the parasite folate metabolism, and are only effective against tachyzoites [5]. Unfortunately, the side effects of these drugs are severe: about 40% of patients are forced to interrupt the treatment because they cannot tolerate it [6]. At the same time, prolonged treatment is necessary for immunocompromised patients [7]. Long-term treatment, side effects and resistance to drugs limit the use of these drugs. Therefore, it is necessary to find high-efficiency and low-toxic drugs in the treatment of toxoplasmosis.

Numerous studies have focused on screening new synthetic or semi-synthetic drugs for treating toxoplasmosis based on plant sources [8]. It is reported that only about 15% of plants have been explored. For instance, *Aloe vera* and *Eucalyptus* methanolic extracts not only have antimicrobial and antioxidant activities, but can also significantly inhibit the growth of *T. gondii* tachyzoites [9].

Studies have indicated that the essential oils from different plants exhibit low cytotoxicity and better inhibition of the proliferation of *T. gondii* [10,11]. Screening antiparasitic drugs from natural products from medicinal plants is an important procedure during new drug research and development [12]. Exploring the mechanism of action of drugs, and then modifying the drugs, obtaining derivatives with better efficiency and low toxicity, is a common way of developing new drugs.

Artemether is a sesquiterpene lactone compound, modified from artemisinin while retaining the peroxide bridge structure, and usually used in combination with artemisinin as a first-line agent in the treatment of malaria [13]. Coartem, composed of a mixture of artemether and lumefantrine, was approved to treat acute and uncomplicated malaria infection in adults and specific children by the US FDA in 2009 [14]. Many studies have indicated that artemisinin derivatives have antiviral, antifungal, anticancer and antidiabetic activities [15,16,17]. Lots of evidence indicates that artemether has potential therapeutic effects on several malignancies, such as glioma, liver cancer and colon cancer [18]. Early in 1990, the antimalarial compound artemisinin and six derivatives were tested for their ability to inhibit *T. gondii* in fibroblasts. Artemether, especially, was the most effective drug [19]. Pretreatment of free tachyzoites with artemether, followed by Vero infection, greatly reduced the concentration of the artemether that killed infected *T. gondii* and, even at the highest concentrations, only produced low cytotoxicity on Vero and J774 cell lines [20]. A previous study revealed that dihydroartemisinin modulated the immune system by promoting CD8+ T lymphocytes and inhibiting B cell responses, suggesting artemisinin and its derivatives can actively modulate the immune system and directly benefit the host [21]. 

The structure of artemisinin and its derivatives contains a unique endoperoxide bridge, which has biological activities [22]. Artemether, with its unique internal structure, can react with ferrous ions to generate freedom, thus causing cell damage. Ooko et al. found that a variety of artemisinin derivatives were associated with 20 iron-related genes in 59 cell lines, including transferrin and transferrin receptor genes, proving that artemisinin derivatives are closely related to iron [23]. The effect of artemisinin is mediated by free iron, which is cleaved in the presence of excessive accumulation of ferrous iron-dependent lipid peroxides called ferroptosis. Inducing mitochondria destruction decreased membrane potential, and accumulated reactive oxygen species (ROS) further altered mitochondrial function [24]. Under these pressure conditions, the production of ROS causes photooxidative damage and cell metabolism disorders [25]. 

Several studies have demonstrated that the biological effects of artemether are related to the regulation of mitochondria [26,27,28]. As an important organelle in eukaryotic cells, mitochondria are involved in plenty of key biological processes, such as cellular respiration, redox reaction and thermogenesis. It is very important for most protozoans, including *T. gondii* [29]. In recent years, studies have shown that *Toxoplasma* infection is associated with neurological disorders. Patients with toxoplasmosis seropositive schizophrenia can be treated with artemether during their first episode [30]. Researchers have also found that artemisinin and its derivatives have a good inhibitory effect on the growth of *T. gondii* in fibroblasts [31].

In our study, artemether was selected to evaluate the inhibitory effect on *T. gondii* in vitro and the detailed mechanism was further studied.

## 2. Materials and Methods 

### 2.1. Host Cells and Parasite Culture

Human foreskin fibroblast (HFF) cells were cultured in Nunc Easyflasks (Thermo Scientific, Waltham, MA, USA), maintained in Dulbecco’s modified Eagle´s medium (DMEM, Thermo Scientific, Waltham, MA, USA), and supplemented with 10% FBS, 100 U/mL penicillin and 10 mg/mL streptomycin at 37 °C in a 5% CO_2_ atmosphere. *T. gondii* RH strain expressing tomato red fluorescent protein (RH-RFP) was constructed under tubulin promoter according to Striepen et al. [32]. RH-RFP was cultured in monolayers of HFF cells. To harvest tachyzoites, heavily infected cells were scraped and passed through a 27-gauge needle, 3–5 times. Cell debris was removed by a 3 µm membrane filter (Whatman, Maidstone, UK). Tachyzoites were counted using a hemocytometer for further experiments.

### 2.2. Drug Compounds

Artemether was purchased from Aladdin Reagent limited company, Shanghai, China. The powder was then dissolved in dimethyl sulfoxide (DMSO) with an initial concentration of 1 M. The solution was stored at −20 °C. The different concentrations of artemether were diluted with DMEM medium. The final concentration of DMSO in the samples used in the experiment was lower than 1% (*v*/*v*). 

### 2.3. Cytotoxicity Assay

The cytotoxicity of artemether was determined in an HFF cell with a CellTiter 96^®^ AQueous One Solution Cell Proliferation Assay (Promega Corp., Madison, WI, USA), according to the manufacturer’s instructions. A total of 1 × 10^5^ cells were seeded in 96-well plates and cultured at 37 °C, in an atmosphere containing 5% CO_2_, for 24 h. They were then treated with different concentrations of artemether (10,000 μM, 2000 μM, 1000 μM, 500 μM, 200 μM and 50 μM) in DMEM complete culture medium. After incubating for 24 h, adding 10 μL MTT solution to each well and incubating for 4 h, 200 μL of DMSO were added to dissolve the formazan crystals. The absorbance of the suspension was measured at 490 nm using an iMark^TM^ Microplate Absorbance Reader (BioRad, Hercules, CA, USA). The 50% cytotoxic concentration (CC_50_) of artemether was calculated using Graphpad Prism 8.0. The cytotoxicity experiment was performed in triplicate, using three separate plates.

### 2.4. Anti-T. gondii Activity of Artemether Evaluated by a Plaque Assay

A confluent monolayer of HFF cells was infected with 5 × 10^2^ tachyzoites for 4 h in 6-well plates. Then, the extracellular parasites were removed, and a fresh medium containing different concentrations of artemether or 0.5% DMSO (vehicle control) was added to each well. Uninfected and untreated wells were used as controls. They were then incubated at 37 °C with a 5% CO_2_ atmosphere for 7 days without any movement. HFF cells were washed with PBS, fixed with 4% paraformaldehyde and stained with 2% crystal violet for 30 min. Finally, the field of vision was randomly selected and photographed under the microscope. The plaque areas were analyzed.

### 2.5. Effects of Artemether on Intracellular T. gondii

Monolayer HFF cells were incubated in a 24-well plate, and RH-RFP tachyzoites were added at a multiplicity of infection (MOI) of 1 and allowed to settle at 37 °C with an atmosphere containing 5% CO_2_ for 24 h. The medium containing extracellular parasites was removed and a fresh medium containing either artemether (31.25 μM, 15.63 μM, 7.81 μM, 3.91 μM and 1.95 μM), 0.3% DMSO (vehicle control) or pyrimethamine (positive control) was added, respectively. After incubating at 37 °C for 24 h, a fluorescence microscope was used to examine the growth of RH-RFP, and the growth rate was statistically analyzed using Image-Pro Express.

### 2.6. Invasion Assay

Invasion experiments were performed as described by Lim SSY et al. [33] In brief, HFF cells were cultured in a 6-well plate, and 3 mL DMEM with 2% FBS were added to each well. Then, 1 × 10^5^ RH and 10 μM artemether were added simultaneously to the wells, respectively, incubating for 20 min, 40 min or 60 min. The supernatant was gently removed, cells were fixed with 2 mL 4% paraformaldehyde for 10 min, washed three times with PBS, blocked by 5% BSA in PBS (BSA/PBS) for 1 h, and washed three times with PBS. They were then incubated with mouse anti-*T. gondii* SAG1 at room temperature for 2 h, followed by Alexa Fluor 488 goat anti-mouse secondary antibodies for 2 h, then washed three times andpermeabilized with 0.1% Triton X-100/PBS for 1 h. Cells were stained with rabbit anti-*T. gondii* polyclonal antibodies followed by Alexa Fluor 594 goat anti-rabbit secondary antibodies. Nuclei were stained with DAPI for 1 h. Five visual fields were randomly selected for observation under the 40× objective of the fluorescence microscope and the parasites in each field were counted. Three repetitions were performed to increase the accuracy of the experiment. The difference between the tachyzoites of the two colors is termed the absolute invasion number of tachyzoites. The ratio of the invasion number to the total number of tachyzoites is termed the invasion rate of tachyzoites.

### 2.7. Intracellular Proliferation Assay

Freshly egressed tachyzoites were allowed to infect HFF monolayers for 2 h, then the medium was changed with containing artemether (30 μM, 15 μM, 7.35 μM, 4.15 μM and 1.37 μM), and 0.2% DMSO, respectively. Invaded parasites were co-cultured at 37 °C in a 5% CO_2_ atmosphere for 24 h and 48 h. A total of 100 parasitophorous vacuoles (PV) were randomly selected and the number of parasites in each vacuole was counted.

### 2.8. Measurement of Mitochondrial Membrane Potential of Tachyzoites 

A total of 1 × 10^7^ fresh tachyzoites were incubated in DMEM containing different concentrations ofartemether (12.5 μM or 3.35 μM), or no drug (vehicle control) at 37 °C with a 5% CO_2_ atmosphere for 60 min. Parasites were stained with JC-1 (Solarbio, Beijing, China) according to the manufacturer’s protocol. The samples were analyzed by flow cytometry. Three independent biological experiments and three technical replicates were performed.

### 2.9. Measurement of Reactive Oxygen Species (ROS)

ROS was measured by ROS Assay Kit—Highly Sensitive DCFH-DA (Jiancheng, Nanjing, China) according to the manufacturer’s instructions. Briefly, 1×10^7^ purified tachyzoites were suspended in 0.1 M PBS and mixed with Highly Sensitive DCFH-DA Dye for 30 min, and then artemether (12.5 μM and 3.35 μM), PBS and H_2_O_2_ were added, respectively, incubating for 2 h at 37 °C. Fluorescence acquisition was measured at 485 nm and 530 nm using a multifunctional microplate reader (BioTek Synergy 2, Genomics Co., Ltd., Waltham, MA, USA) 

### 2.10. Statistical Analyses

All data were analyzed using Graphpad Prism 8.0. The differences between treatments and controls were assessed using analysis of variance (one-way and two-way ANOVA) and the results in comparisons between any two groups were considered as differences if *p*  <  0.05. Flow cytometry assay was analyzed using CytExpert 2.3.

## 3. Results

### 3.1. Cytotoxicity of Artemether 

The cytotoxic potential of artemether on HFF cells was confirmed before anti-*T. gondii* activity study. According to the MTT assay result, the concentration of artemether that induced a 50% HFF cell mortality (CC_50_) was 866.4 μM (Figure 1). 

Cytotoxicity was evaluated using a CellTiter 96^®^AQueous One Solution Cell Proliferation Assay. Cell viability was calculated by comparing the treatment group with the negative control group, and the base of log concentrations was used to analyze the CC_50_. All data are presented with error bars, the experiments were performed in triplicate and three technical replicates were included in each experiment.

### 3.2. Antiparasitic Activity of Artemether In Vitro

The antiparasitic effect of artemether was preliminarily evaluated by plaque assay. From the results, we found that the plaques were fewer and smaller in artemether-treated groups (25.25 μM or 7.5 μM), compared to DMSO-treated and untreated groups (Figure 2A,B). These results indicated that artemether could inhibit the parasite proliferation under the safe concentration. As seen in Figure 3, the IC_50_ of artemether is 9.035 μM, this concentration is much lower than the safe concentration (866.4 μM). We could find that when the drug concentration reached 31.25 μM, the antiparasitic effect is similar to that of the pyrimethamine, within this concentration, the inhibition increased in a dose-dependent manner (Figure 4A,B). 

RH-RFP was cultured in monolayers of HFF cells and treated with different concentrations of artemether for 24 h. The growth rate of RH-RFP tachyzoites was statistically counted using Image-Pro Express and plotted by Graphpad Prism 8.0, and the base of log concentrations were used to analyze the IC_50_. The data are presented as the mean ± SD. Three independent experiments were performed, and three technical replicates were included in each experiment.

### 3.3. Effect of Artemether on the T. gondii Invasion

The invasion ability of *T. gondii* is an important prerequisite for its proliferation in different hosts. To test whether the artemether inhibited *T. gondii* proliferation in HFF by inhibiting invasion, the related assay was performed. From the results, we could find that tachyzoites were treated with 10 μM artemether for 20, 40 and 60 min, the invasion rates were 20.07%, 36.64% and 48.58%, respectively. In the untreated group, the invasion rates were 31.77%, 53.53% and 66.92% (Figure 5). Although the invasion rate was slightly inhibited by artemether (*p* < 0.05), the differences were not significant, which indicated that the artemether-inhibited *T. gondii* proliferation was not related to the invasion.

### 3.4. Inhibition of Artemether on T. gondii Intracellular Proliferation 

The inhibition of artemether on *T. gondii* intracellular proliferation was evaluated by replication assays on HFF cells. One hundred PV were randomly selected to count the tachyzoite number in each group. As shown in Figure 6A, we could find that nearly 80% of PVs contained one or two tachyzoites in the 30 μM artemether group, which indicated that the proliferation rate was significantly inhibited by artemether. Compared to the control group, artemether-treated groups showed a significantly reduced proliferation rate in a dose-dependent manner treated by artemether for 48 h (Figure 6A). Similar results were found when tachyzoites were treated for 24 h (Figure 6B).

### 3.5. Artemether Impaired Mitochondrial Membrane Potential of T. gondii

To investigate the potential anti-*Toxoplasma* mechanism of the artemether, tachyzoites were treated with different concentrations of artemether. From the results, we could find that 61.91% and 31.23% of mitochondrial membranes were damaged after treatment by 12.5 μM and 3.35 μM artemether, respectively (Figure 7A). According to the mean fluorescence intensity, the mitochondrial membrane potential in the 12.5 μM artemether group was significantly lower than the control group (*p* < 0.01) (Figure 7B). This result suggested that artemether may exert antiparasitic effects by disrupting the mitochondrial membrane of *T*. *gondii*.

### 3.6. Artemether Increased ROS Production of T. gondii

Since the mitochondrial membrane potential of *T. gondii* was changed, we wanted to further analyze the changes in ROS in extracellular tachyzoites. As shown in Figure 8, we found that artemether could increase ROS activity of extracellular tachyzoites, especially in the 12.5 μM artemether-treated group, ROS activity was significantly higher compared to that in the control group (*p* < 0.001) (Figure 8). From this result, we speculated that the antiparasitic activity of artemether is associated with the induction of ROS accumulation in *Toxoplasma*. 

## 4. Discussion

Toxoplasmosis is one of the most challenging protozoan diseases in public health. In the absence of a safe and effective vaccine or drug to eradicate toxoplasmosis, the disease remains one of the great global challenges [34]. Hence, the development of new therapeutic drugs is urgent in the treatment of toxoplasmosis. Chinese medicine from natural plants is widely used to control different parasites due to their safety and effectiveness. In recent years, many researchers focused on screening anti-*Toxoplasma* drugs from natural plant extracts, such as plant essential oils and other products [35,36]. Numerous studies have shown that artemisinin and its derivatives not only effectively treat malaria, but also inhibit the proliferation of cancer cells and promote their apoptosis and ferroptosis [37,38,39,40]. 

In this study, we systematically studied the effect of artemether on *T. gondii* inhibition in HFF cells and explored its potential inhibition mechanism. We found that the IC_50_ of artemether was 9.035 μM (Figure 3), while other studies had different results, such as one study indicating the EC_50_ of artemether was 0.286 μM and another showing the IC_50_ was14.64 μM [20,41]. This difference might be caused by the drugs coming from different companies, different solvents and host cells. The anti-*Toxoplasma* effect of artemether was further assessed using the growth assay, and inhibition showed a dose-dependent manner (Figure 4). It is well known that the lysis cycle of *T. gondii* includes the processes of adhesion and invasion of host cells, the exponential growth of 2^n^ in intracellular proliferation, and the escape to find a new host after the disintegration of the host cells. To study the inhibited mechanism, an invasion assay was first carried on. We found that artemether did not significantly inhibit the *T. gondii* invasion, although the invasion rate was slightly influenced by it. Intracellular proliferation was then analyzed and we happily found that PVs contained fewer tachyzoites in the artemether-treated group, compared to the untreated group, which indicated artemether inhibited the *T. gondii* intracellular proliferation. Shaw et al. proved that once *T. gondii* enters the host cell, it will stop moving and proliferate in the way of the daughter cell budding [42]. Parts of the host’s mitochondria and endoplasmic reticulum are recruited to form PV, allowing *T. gondii* to proliferate. The specific molecular mechanism of artemether inhibiting *T. gondii* intracellular replication is unclear and needs further investigation. 

The mitochondrial membrane potential is often detected with a JC-1 fluorescent probe to analyze mitochondrial configuration and function [43]. The ratio of monomers to aggregates is often used to measure the proportion of mitochondrial depolarization. Mitochondria are “little power stations”, they are important organelles that provide energy to host cells. The mitochondrial electron transport chain (ETC) is an essential pathway for providing energy and participating in the oxidation–reduction process of *T. gondii* [44]. Mitochondrial disorders lead to disruptions in metabolism, which in turn release ROS [45]. More evidence indicates that intracellular ROS production is mainly due to changes in mitochondria, in which either NADH or FADH_2_ react with O_2_ arising from more ROS at complexes I and III of the electron transport chain (ETC) [46]. The production of intracellular ROS leads to disturbances in mitochondrial DNA, lipids and protein synthesis [47]. 

In this study, we explored the relationship between the mechanism of artemether against *T. gondii* and mitochondrial membrane potential, and then explored its association with ROS release. Flow analysis revealed that the mitochondria membrane potential of *T. gondii* was reduced and ROS was increased when treated with artemether. These results suggested that *Toxoplasma* was inhibited due to oxidative stress treated by artemether. 

Cytotoxicity of artemisinin and its derivatives to cancer cells are thought to involve the generation of oxidative stress, following cleavage of the endoperoxide bridge. Heme synthesis can modulate artemisinin cytotoxicity towards cancer cells [48]. Additionally, artemisinin and artesunate were found to exert anti-malarial effects by destroying the basic chemical of an endoperoxide bridge that generates carbon-centered free radicals which increase oxidative stress and arbitrarily modify molecular structure such as lipid, protein and DNA damage via alkylation [49]. Taking the anti-*plasmodium* principle as a reference, whether artemether acts against *T. gondii* by breaking the intramolecular peroxide bridges to activate divalent iron or heme iron needs further study. 

## 5. Conclusions

In conclusion, artemether inhibited *T. gondii* intracellular proliferation in a dose-dependent manner *in vitro*. The results speculated that the mechanism of artemether against *T. gondii* was related to the change in mitochondrial membrane and the increase in ROS production, which provided a theoretical basis for optimizing artemether derivatives and further improving their anti-*Toxoplasma* efficacy, and the results were expected to bring new hope for the treatment of toxoplasmosis.

## Figures and Tables

**Figure 1 metabolites-13-00476-f001:**
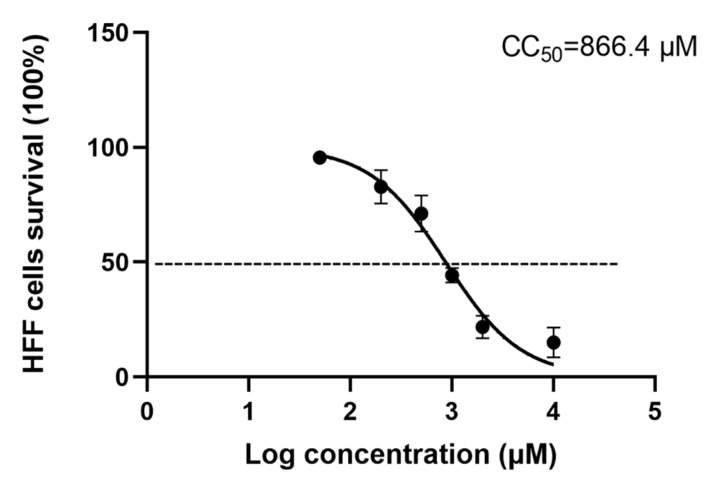
Cytotoxicity assay of artemether on HFF cells.

**Figure 2 metabolites-13-00476-f002:**
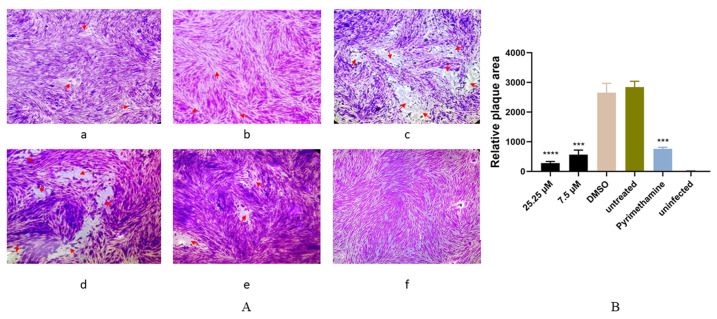
Plaque test for preliminary detection of the anti-*T. gondii* activity of artemether. *T. gondii* plaques were observed and counted when cells were treated with different concentrations of artemether for 7 days. (**A**): (**a**,**b**) HFF cells infected by *T. gondii* and treated with artemether of 25.25 μM or 7.5 μM. (**c**,**d**) HFF cells infected by *T. gondii* and treated with DMSO or untreated. (**e**) HFF cells infected and treated with pyrimethamine as positive control. (**f**) HFF cells were not infected and untreated. (**B**): The relative areas of plaque in each group were calculated. All the above experiments were repeated three times and 3 technical replicates were included in each experiment. Red arrows (*T. gondii* plaques) *** *p* < 0.001, **** *p* < 0.0001.

**Figure 3 metabolites-13-00476-f003:**
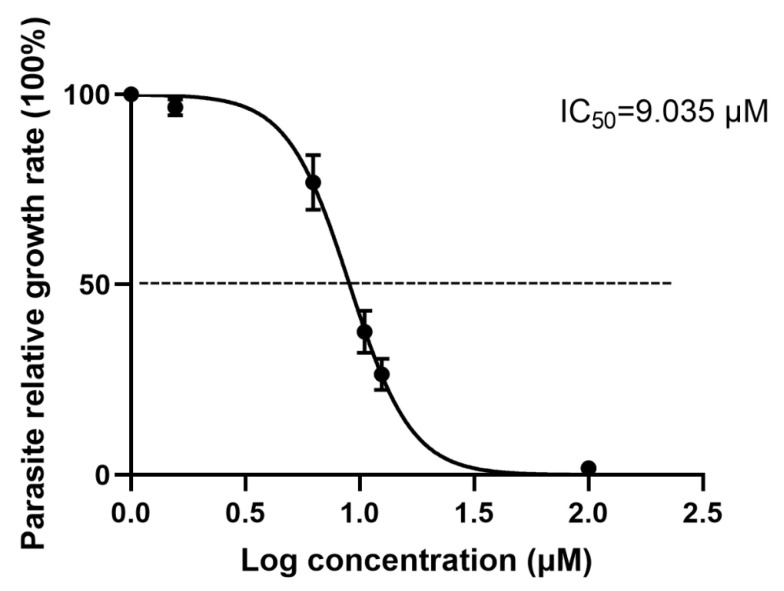
The 50% inhibition concentrations (IC_50_) of artemether.

**Figure 4 metabolites-13-00476-f004:**
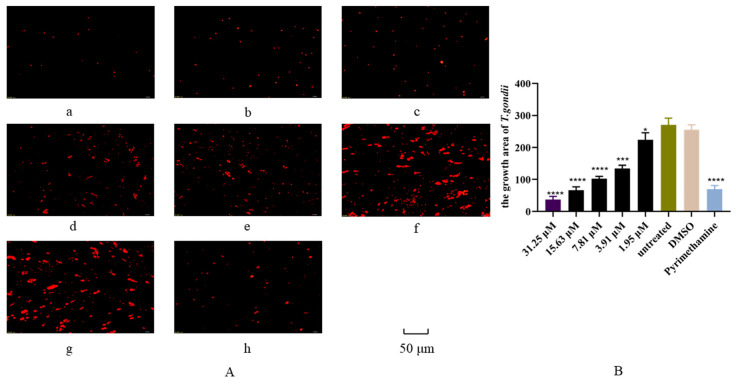
Anti-*T. gondii* activity of artemether was evaluated by RH-RFP growth assay. The HFF cells were infected by RH-RFP and treated with different concentrations of artemether and pyrimethamine for 24 h, respectively. The growth rate was calculated by measuring the fluorescence intensity. (**A**): Fluorescence area indicates the growth rate of *T. gondii*. (**a**–**e**) HFF cells were infected with RH-RFP and treated with different concentrations of artemether. (**a**) 31.25 μM, (**b**) 15.63 μM, (**c**) 7.81 μM, (**d**) 3.91 μM, (**e**) 1.95 μM, (**f**) untreated, (**g**) DMSO and (**h**) pyrimethamine as positive control. (**B**): Statistical analysis of the growth rates in different treated groups. Each bar presented as the mean± standard error, three independent experiments were performed, and 3 technical replicates were included in each experiment. All images were observed under 200× magnification and scale bars equal to 50 μm. ** p* < 0.05, **** p* < 0.001, ***** p* < 0.0001.

**Figure 5 metabolites-13-00476-f005:**
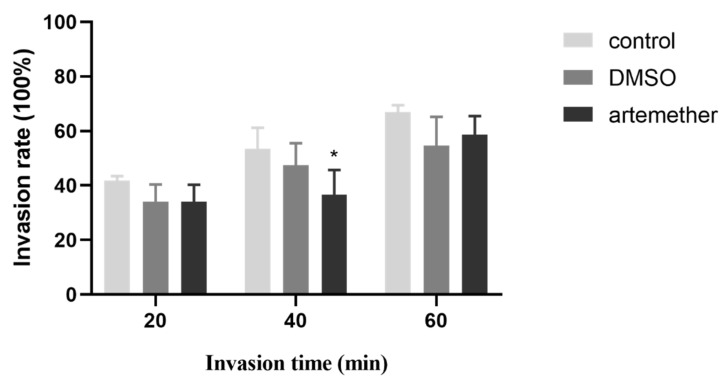
Effect of artemether on the invasion of *T. gondii* was evaluated by two immunofluorescent dyes. *T. gondii* invasion rate was evaluated using two immunofluorescent dyes. RH tachyzoites were treated with artemether for 20, 40 and 60 min, respectively. The ratio of the invasion number to the total number of tachyzoites is termed the invasion rate. Three independent experiments were performed and three technical replicates were included in each experiment. ** p* < 0.05.

**Figure 6 metabolites-13-00476-f006:**
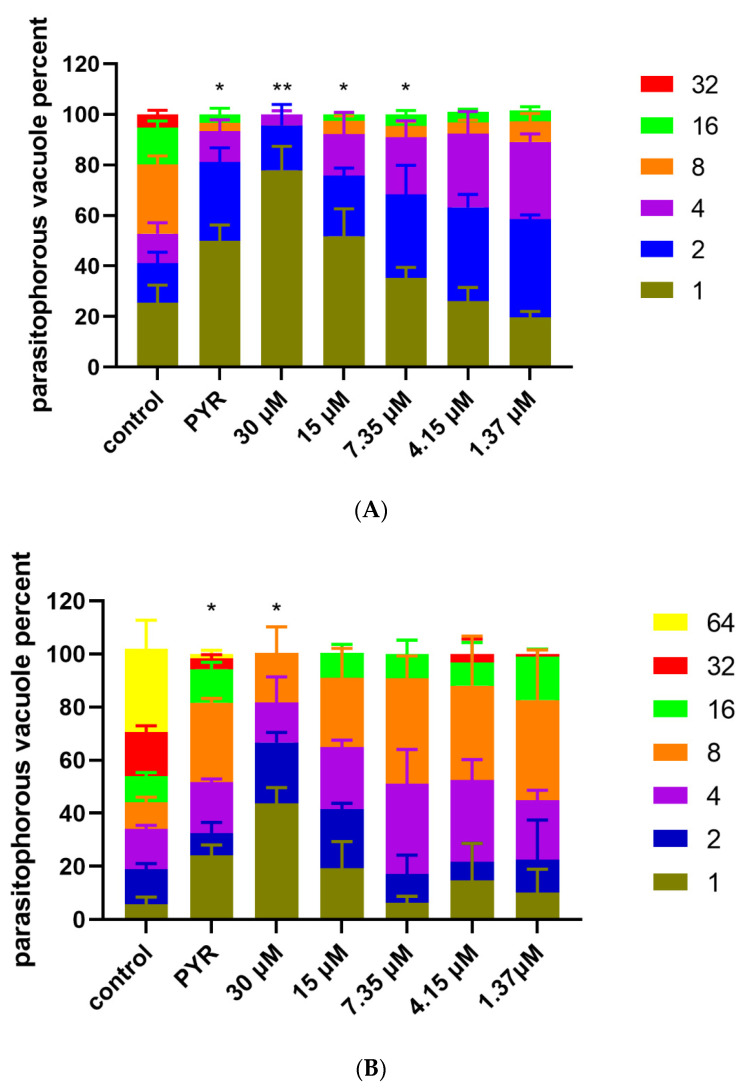
Effect of artemether on the *T. gondii* intracellular proliferation. The intracellular proliferation was examined after *T. gondii* tachyzoites were treated with different concentrations of artemether for 48 h (**A**) and 24 h (**B**). 1, 2, 4, 8, 16, 32 and 64 represent the number of tachyzoites in each PV. The experiments were performed in triplicate, and 3 technical replicates were included in each experiment. Significant differences compared to the control are indicated by ** p* < 0.05, *** p* < 0.01.

**Figure 7 metabolites-13-00476-f007:**
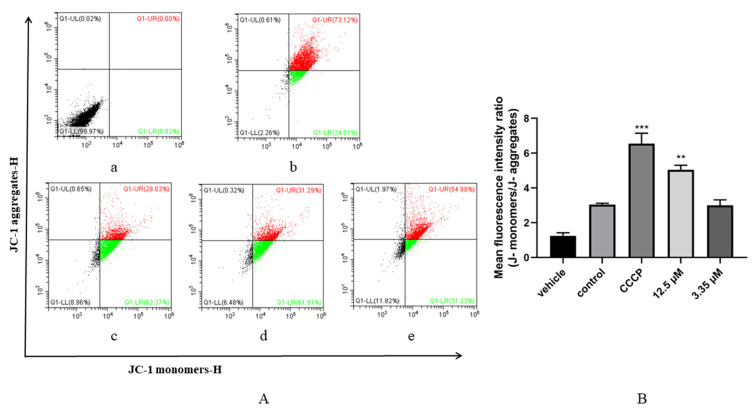
The change in *T. gondii* mitochondrial membrane potential. (**A**): Changes in red and green fluorescence were used to evaluate the mitochondrial membrane potential of extracellular *T. gondii*. The proportion of tachyzoites in quadrant 2 (Q1−UR) represents the degree of depolarization of mitochondrial membranes of *T. gondii*. (**a**) extracellular tachyzoites were treated with PBS, (**b**) extracellular tachyzoites were added using a JC−1 probe with PBS, (**c**) extracellular tachyzoites were treated with CCCP as positive control, (**d**,**e**) tachyzoites were treated with artemether of 12.5 μM and 3.35 μM, respectively. (**B**): Statistical analysis of fluorescence intensity. The data were represented by mean ± SD of three experiments, and each experiment was conducted in triplicate, and 3 technical replicates were included in each experiment. Significant differences compared to the control are indicated by *** p* < 0.01, **** p* < 0.001.

**Figure 8 metabolites-13-00476-f008:**
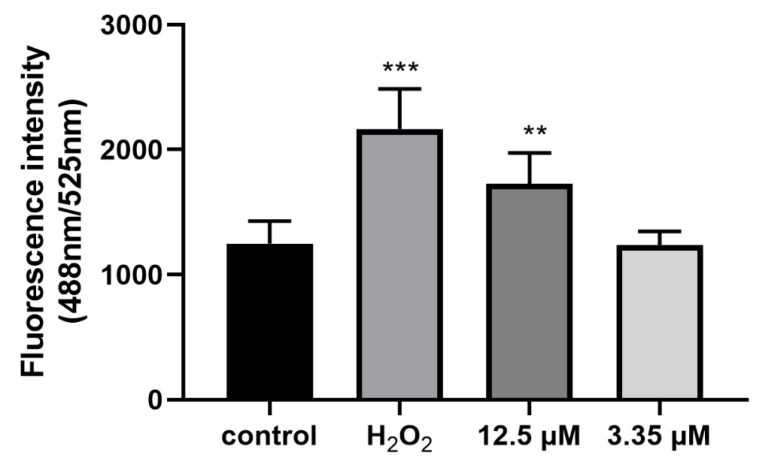
Production of reactive oxygen species (ROS) in *T. gondii.* Extracellular *T. gondii* tachyzoites were pretreated with different concentrations of artemether (12.5 μM and 3.35 μM), PBS (negative control) and 100 mM H_2_O_2_ (positive group). Each bar represents the means ± SD of three different experiments performed in triplicate, and three technical replicates were included in each experiment. The results were analyzed using one-way ANOVA. *** p* < 0.01, **** p* < 0.001.

## Data Availability

Not applicable.

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
