# Peer review of "In Vitro Evaluation Reveals Effect and Mechanism of Artemether against Toxoplasma gondii"

_metabolites, 2023, doi:10.3390/metabo13040476_

Round 1

Reviewer 1 Report

In the present study, the authors have evaluated the effect and potential mechanism of Anti-Toxoplasma gondii activity induced by Artemether.

1.     The lines 12-15 in the abstract Is not clear and the authors need to reframe the sentences.

2.     Authors should define the details of the construct design for Type I strain Tachyzoites expressing RFP. 

3.     Authors should give details about the number of biological and technical replicates included for each experiment and the data should reflect in their respective figures.

4.     Authors should clarify the base of log concentrations used.

5.     Authors should provide more clear photos of the plaques and point out the differences seen among different groups for better understanding. 

6.     Authors have found less and smaller plaques in Artemether treated group. It would be good if the authors can provide the data on the number of plaques seen among different groups.

7.     Authors should provide detailed description in each figure legends.

8.     The authors should provide labels and description of the scales for the fluorescent images in this paper. 

9.     The authors should provide the details on the numbers of PV counted for each replicate.

10.  The authors must explain and discuss their results in more details.

11.  No references can be found.

12.  English needs improvement and the font style used is not the same style throughout the manuscript.

Reviewer 2 Report

Qiong Xu et al. are reporting an interesting manuscript on the mechanism study of artemether agonist T. gondii. A set of cell-based assays including cytotoxicity assay, plaque assay, invasion assay, Intracellular proliferation assay, evaluation of the ROS, etc. were conducted. Overall, the research was well-organized, and the manuscript was well-written. The reviewer would suggest a minor revision if the following comments can be addressed.  

1.      Figure 4A uses fluorescence to visualize the growth during the treatment with different concentrations. Is it possible to change the background color when plotting? Or is there a better for illustration? Current coloring makes subplots a-f hard to read. 

2.      In vitro assays are comprehensive to (1) show the dose-dependent inhibition of T. gondii intracellular proliferation, and (2) suggest a mechanism of increasing of ROS production and changing mitochondrial membrane. The reviewer is wondering if authors can continue the discussion of the mechanism towards the molecular level by indicating specific target(s) that can be regulated by artemether. 

3.      The English language is fine. Some minor styling and spelling modifications can be beneficial. 

Reviewer 3 Report

The manuscript presents the results of research on the elucidation of the mechanism of biological activity of the active pharmaceutical ingredient artemether. The obtained results indicated the effect of this drug on the change of mitochondrial membrane and the increase of ROS production. These likely changes are responsible for the activity of artemether against Toxoplasma gondii. The work is worth publishing, but after the authors have clarified the issues listed below.

 There are no References in this manuscript. I do not know what literature items and from what years the authors of the work cite. This significantly hinders a reliable assessment of the content of the manuscript.

 Artemether is an organic synthetic drug, a derivative of artemisinin, which is of plant origin, it was obtained from Artemisia annua. Because there are doubts about the manufacturing process of artemisinin derivatives and their brainstem toxicity, the US FDA did not approve this drug for use. Are the authors aware of information about the suspicion of a similar operation of the artemether. If so, this information should be included in the manuscript.

 The first two paragraphs of section 4. Discussion (lines 312-340) should be moved to section 1. Introduction. They provide good justifications for taking up the research carried out by the authors.

The meaning of many of the abbreviations in the manuscript is not explained, e.g. DMSO, MOI, PV.

Round 2

Reviewer 1 Report

No further comments